# Production of Polyhydroxyalkanoates for Biodegradable Food Packaging Applications Using *Haloferax mediterranei* and Agrifood Wastes

**DOI:** 10.3390/foods13060950

**Published:** 2024-03-20

**Authors:** Lorena Atarés, Amparo Chiralt, Chelo González-Martínez, Maria Vargas

**Affiliations:** Instituto de Ingeniería de Alimentos—FoodUPV, Universtitat Politècnica de València, 46022 Valencia, Spain; loathue@tal.upv.es (L.A.); dchiralt@upv.es (A.C.); cgonza@tal.upv.es (C.G.-M.)

**Keywords:** polyhydroxyalkanoate, biodegradable polymers, food packaging, waste valorization, *Haloferax mediterranei*

## Abstract

Polyhydroxyalkanoates (PHAs) are high-value biodegradable polyesters with thermoplastic properties used in the manufacturing of different products such as packaging films. PHAs have gained much attention from researchers and industry because of their biobased nature and appropriate features, similar to conventional synthetic plastics. This review aims to discuss some of the recent solutions to challenges associated with PHA production. The implementation of a cost-effective process is presented by following different strategies, such as the use of inexpensive carbon sources, the selection of high-producing microorganisms, and the functionalization of the final materials to make them suitable for food packaging applications, among others. Research efforts are needed to improve the economic viability of PHA production at a large scale. *Haloferax mediterranei* is a promising producer of PHAs due to its ability to grow in non-sterile conditions and the possibility of using seawater to prepare the growth medium. Additionally, downstream processing for PHA extraction can be simplified by treating the *H. mediterranei* cells with pure water. Further research should focus on the optimization of the recycling conditions for the effluents and on the economic viability of the side streams reutilization and desalinization as an integrated part of PHA biotechnological production.

## 1. Introduction

Products based on conventional fossil fuel-derived plastics are important commodities with wide applications in various fields such as household, food industry, medicine, packaging, and transport [1]. Conventional plastics are cheap to produce, light, resistant, and easily molded as per requirement [2]. However, they also show important disadvantages, such as their non-biodegradable nature, non-sustainable origin, and release of toxic materials into the environment for a prolonged time [3]. The increasing global plastic consumption has given rise to plastic accumulation in the environment, including both land and oceans [4]. Moreover, an important proportion of this problem is attributable to the food industry, given that conventional plastics have been massively used as food packaging given their good properties and low cost. In fact, in 2021, packaging was the largest world plastic market, with polyethylene (PE), polypropylene (PP), and polyethylene terephthalate (PET) as the most used polymers [5]. 

The problem of plastic pollution urges research efforts to find sustainable alternatives such as biodegradable plastics of renewable origin. In this sense, biodegradable plastics emerge as substitutes for conventional ones, and among them, microbial-origin polyhydroxyalkanoates (PHAs) are considered one of the best alternatives [1,3]). Contrarily to conventional petroleum-based synthetic polymers, PHAs are completely biosynthetic and biodegradable under different environments (soil, compost, and aquatic media), have zero toxic waste, and are completely recyclable into organic waste [4]. The varying structures of PHAs have allowed the development of a number of applications in place of petroleum-derived plastics. These include fibers, biocompatible implants, controlled drug delivery systems, biofuels, and biodegradable plastics for packaging [4]. Due to their high biocompatibility and non-toxicity, PHAs have been used mainly for biomedical applications such as sutures and tissue engineering [6]. Given the good properties of PHAs as packaging materials, there is also increasing demand for these for disposable packaging goods such as antimicrobial food containers [7]. PHA-based packaging materials are able to extend the shelf-life of packaged foods while reducing the environmental impact [8]. PHAs also show an important benefit in comparison with other biobased polymers, such as polylactic acid (PLA), as PLA is not biodegradable in aquatic environments.

In spite of all the advantages described, PHAs are still too expensive to be produced at an industrial scale, and therefore, they are not yet competitive in the plastics market, where only 1.5% of world plastic production corresponds to biobased/bio-attributed plastics [5]. Important research efforts are being made to design efficient and economically viable bioprocesses to produce these biopolymers for diverse applications, including food packaging. This review provides a state-of-the-art overview of the most recent advances in PHA production using *Haloferax mediterranei* and inexpensive carbon sources such as agrifood waste. Moreover, a wide overview of different strategies for PHA competitivity improvement in the market of plastics for food packaging has been included.

## 2. Structure, Properties, and Degradation Mechanism of PHAs

PHAs, currently used mostly for biomedical applications, have attracted attention as potential materials for food packaging because of their appropriate features, similar to synthetic plastics traditionally used for this purpose [9]. These biodegradable polymers are considered good substitutes for petrochemical polymers such as polyethylene, polypropylene, nylon, and polyvinylchloride [2]. Due to their hydrophobic nature, they can be used to be in contact with high-moisture food products while maintaining their structure. They are a natural feedstock of renewable origin, biodegradable under a wide range of environmental conditions, compostable, and processable with conventional plastics machinery [10]. 

PHAs are linear polyesters of 3-hydroxy fatty acid monomers, where an ester bond ties the carboxyl group and the hydroxyl group of two repeating units. Based on the number of carbon atoms in the side chains, they are classified into (1) short chain length PHA (3–5 carbon atoms); (2) medium chain length (6–14 carbon atoms); and (3) long chain length (15 or more carbon atoms) [3]. A wide variety of PHAs, comprising as many as 150 distinct monomer units, have been studied [9]. The two most investigated types of PHAs are poly(3-hydroxybutyrate) (PHB, the most frequently occurring PHA) and poly(3-hydroxybutyrate-co-3-hydroxyvalerate) (PHBV). Figure 1 shows the general structure of PHAs, as well as those of PHB and PHBV.

In PHB, there are four carbon atoms in the monomer, and the R-chain is a methyl group. PHB is difficult to process due to its stiffness and brittleness because of its high degree of crystallinity and the closeness between its fusion and decomposition temperatures (175 °C and 185 °C, respectively) [4]. The introduction of a 3-hydroxyvalerate (3HV) monomeric unit disrupts the crystallinity of PHB, thus yielding PHBV. As compared with PHB, PHBV has been reported to have lower crystallinity, T_m_ and T_g_, while also biodegrading quicker [1,11,12,13]. Furthermore, an increase in the content of 3HV monomeric unit in the copolymer improves the impact resistance, toughness, elasticity, flexibility, and overall mechanical and thermoplastic properties of PHBV, thereby making these similar to those of fossil-derived polymers [14,15]. For these reasons, PHBV is regarded as a promising polymer for commercial production and application and has gained much attention from both academia and industry [15,16].

The physical properties of PHAs can be equivalent to those of conventional synthetic plastics due to the wide chemical variety of their radicals. Depending on the monomer composition, PHA can be yielded from rigid and brittle plastic materials to soft elastomers, rubbers, and adhesives [17]. 

Some PHA packaging materials offer good resistance to moisture because of their hydrophobic nature [18] and gas barrier properties comparable to polyvinyl chloride (PVC) and polyethylene terephthalate (PET) [17]. Compared with polysaccharide-based materials, PHA films exhibit much better moisture barrier properties, whereas the gas barrier capacity is inferior [19].

Table 1 summarizes the range of typical properties of polyhydroxyalkanoate (PHAs) as affected by the production technique (solvent casting or melt blending). 

PHB homopolymer-based materials are brittle and stiff, with poor impact resistance, due to their high degree of crystallinity and the secondary crystallization that occurs after film processing with aging [18,29]). As compared with PHB, PHBV-based packaging materials exhibit improved stretchability and tensile strength as the content in the copolymer increases [29,32]), since 3-hydroxyvalerate units hinder the formation of hydroxybutyrate crystals [33].

As shown in Table 1, the properties of PHA-based packaging films are also known to be dependent on the processing technique (melt blending or solvent casting). In this sense, PHBV–PHBV bilayer films obtained by thermoprocessing exhibited greater mechanical resistance and lower water vapor permeability (WVP) and transparency, as compared with the corresponding multilayer obtained by casting [28]. In previous studies, PHB films prepared by solvent casting show higher elongation at break and greater impact resistance than those obtained by melt blending and compression molding, probably due to the finer spherulitic morphology associated with the low crystallization temperature, which results in high nucleation density [34]. Similar results have been observed for copolymers, such as poly(3-hydroxybutyrate-co-3-hydroxyhexanoate) (P(3HB-3HH) or PHBH), where the solvent-casted films were more stretchable than those obtained by compression molding [29]. Regarding thermal properties, PHB solvent-casted films exhibit lower glass transition temperature (T_g_) and melting temperature (T_m_) values than those obtained by melt blending and thermo-compression. However, the film barrier capacities are in the same order of magnitude for PHB films obtained either by solvent casting or thermoprocessing.

The biodegradation of PHAs is known to primarily occur through surface erosion via enzyme-catalyzed hydrolysis [35,36], gradually spreading to the inner bulk, which remains usually better preserved. Thus, the biodegradation process starts with the adhesion of microorganisms to the film surface (biofilm formation); this is reported to be the rate-limiting step in the biodegradation of PHAs. After biofilm formation, extracellular depolymerase catalyzes the hydrolytic bond cleavage of the polymer, leading to the formation of lower molecular compounds that can be bioassimilated, leading place to an increment of the cell biomass and generation of compounds such as CO_2_ and water.

## 3. Production of PHAs

PHAs are naturally synthesized by a wide range of prokaryotic microorganisms, both bacteria and archaea [8]. *Cupriavidus necator*, *Ralstonia eutropha*, *Escherichia coli*, as well as bacterial strains of *Pseudomonas* sp. (*Pseudomonas putida*, *Pseudomonas oleovorans*) and *Bacillus* sp. (*Bacillus megaterium*) are among the more commonly used microorganism for PHA biosynthesis [15]. Both cultivations of a single strain or mixed microbial consortia can be used for PHA production [1]. In fact, the possibility of PHA commercialization might be enhanced using mixed microbial cultures, and therefore, increasing efforts are being made to optimize the application of consortia to PHA biosynthesis as an inexpensive alternative to processes based on pure, monoseptic cultures [37,38]. 

The synthesized PHAs accrue as intracellular granules during stressful conditions, i.e., growth with excess carbon source but limited supply of nitrogen or phosphorous, to serve as energy reservoirs [3,8,39]. The polymer synthesis assists the microorganism in adapting to this nutritional scarcity in harsh environmental conditions [3,8,15,39]. Thus, the overall polymer production achieved is dependent upon both the biomass concentration and the accumulation of the polymer in the cytoplasm, which have to be optimized for high volumetric productivity [38]. Among the different regimes of PHA production reported at laboratory and pilot plant scales, two-stage continuous processes are best suited for maximized bioplastic yield. In the first stage, both carbon and the growth-limiting components are continuously supplied for maximized biomass growth. In the second stage, the carbon source is fed continuously, but no more growth-limiting substrate is provided for PHA accumulation. Under specific conditions, PHAs can accumulate up to 90% of the cell dry mass (CDM) [40]. PHA-producing microorganisms can grow on a wide variety of carbon sources, such as pure sugars, glycerol, or volatile fatty acids, among others. 

Even though the production of PHAs has attracted a lot of attention in recent years as ‘‘green plastics” [41], and its production market is increasing quickly, its high production cost is still a barrier to market expansion [1]. The utilization of bacteria that require controlled operation circumstances and special feedstock has made the production cost of PHA several times higher than that of conventional plastics [9]. Koller et al. [42] speculated that the production price of PHBV through a whey-lactose-based process would be €2.82/kg. Garcia et al. [43] stated that the production cost of PHBV through fed-batch fermentation of *Haloferax mediterranei* utilizing renewable resources would be US$6.29/kg. Bhattacharya et al. [39] estimated US$2.05/kg of PHBV when the annual production was simulated as 1890 tons. Hence, the current price of commercial production of PHA is much higher than that of polypropylene plastic, which has been reported to be €1.0 per kg [44] or less than 1 US$/kg [45]. Thus, large research efforts are being made to increase the competitiveness of PHAs in the plastics market.

## 4. Downstream Processing of PHAs

The downstream processing of PHAs plays a significant role in both the quality of the final material and the cost of production. Efficient recovery and purification have been estimated to represent over half of the overall PHA production cost [46,47]). In addition to the profitability of the process, PHA recovery techniques need to be environmentally sustainable, which implies avoiding the use of hazardous solvents and excessive amounts of energy [48]. PHAs accumulate inside the microbial cells, and to recover them from it, the separation of non-PHA cell biomass through cell disintegration is required [49]. The PHA extraction method determines the properties of the product (molecular mass and purity) as well as the recovery yield [48]. This method has to be selected after considering different factors such as the microbial strain, type of PHA, PHA load in biomass, impact on PHA properties, and subsequent application of PHA [50].

The separation of biomass from the broth can be accomplished using techniques such as sedimentation, centrifugation, or flocculation [48,49]. After this, mechanical or chemical methods are applied for cell membrane lysis, such as high-pressure homogenization, ultrasonic digestion, or solvation with halogenated solvents. Solvent-based PHA extraction with halogenated compounds, predominately chloroform, is considered the benchmark for extraction performance [48] because it produces superior recovery yields and maximum product purity. Due to its convenient use, this procedure is the best-established PHA recovery method at the laboratory scale, but the use of noxious halogenated solvents and their disposal entail high environmental and health risks and should not play a role in a sustainable production chain. For this reason, other less hazardous solvents and alternative PHA recovery methods are being developed [48], such as enzymatic methods [51]. Only a few cases of spontaneous liberation on non-halophile microorganisms have been reported [46], in which case, the use of simpler and much cheaper purification processes such as one-step centrifugation and washing with water could be possible. Additionally, cell disruption for improved PHA liberation can be enhanced by previously applying pretreatments such as heating or freezing [45]. After extraction, a purification step could be added to the process to attain higher purity, for which hydrogen peroxide or ozone can be used [45]. In summary, although there are reports on PHA extraction at a laboratory scale, the development of efficient and cost-effective processes at the industrial scale is still needed [49].

Despite the progress being made in the production of PHA, these bioplastics remain commercially unappealing due to challenges associated with high production cost and complex downstream processing, which includes the expensive carbon sources, the requirement of organic substitutes for production of different types of PHA, sterile fermentation conditions, and expensive extraction processes [15]). Aiming to improve the competitiveness of PHAs in the plastics market by both reducing their production cost and increasing their value, different strategies are needed. Some of these will be addressed in the following sections, namely: (1) using inexpensive carbon sources, such as agricultural waste, (2) reducing the cost of downstream processing of PHAs, (3) selecting high-producing microorganisms advantageous for low-cost production and (4) functionalizing the final materials to make them suitable for food packaging applications.

## 5. Production of PHAs from Inexpensive Carbon Sources

The high production cost of PHAs is linked to the use of purified substrates such as glucose, fructose, or propionic acid [52]. In fact, feedstock cost accounts for over 40% of the total annual operating cost of PHA production [53], and therefore, the availability of cheap substrates limits the production of PHA on a large scale [54]. Furthermore, supplementation of 3HV precursors (using propionic or valeric acids) is generally required by producing bacteria in order to specifically produce PHBV, thus contributing to an increase in the production cost.

Aiming to reduce feedstock cost and make the whole process more cost-effective, research efforts have focused on using a wide variety of inexpensive waste feedstocks [1]. These include glycerol from biodiesel industries, lignocellulosic biomass from numerous origins, waste-cooking oil, agro-food wastes (coffee waste, rice bran, fruit waste, etc.), both macro and microalgae, waste paper and effluents of dairy industries fractions of municipal solid waste, and forestry and agricultural residues, as well as woody and herbaceous crops [4,15,49,52].

Agro-food waste contains high concentrations of organic nutrients such as carbohydrates, amino acids, lipids, and phosphates, which can be reintegrated into the production chain. Koller et al. [54] listed the basic requirements of inexpensive substrates for biotechnological production of PHA, which include being available in sufficient amounts at a constant quality over the year, keeping a constant composition, and presenting high resistance against microbial spoilage over storage. Among food waste materials, lignocellulosic biomass has attracted attention in recent years as a very promising substrate for PHA production because it is abundant, cheap, and does not compete with the human food chain [9,55]. Lignocellulose is mainly comprised of cellulose, hemicellulose, and lignin. As can be observed in Figure 2, the hydrolysis of cellulose and hemicellulose yields fermentable sugars, the carbon source for PHA production, and a phenolic-rich fraction. Nevertheless, the enzymatic hydrolysis of native lignocellulose produces a very low fraction of saccharification (less than 20% glucose from the cellulose fraction) [56] because lignin and lignin-derived phenolic compounds inhibit lignocellulosic enzymes. Therefore, the relatively low amounts of sugars and the presence of inhibitors in hydrolysates are important disadvantages to the utilization of lignocellulosic materials as substrates for biotechnological purposes, for which the profitable utilization of lignocellulose-based substrates is a strategic goal [55,57].

Thus, lignocellulosic biomass should be pretreated in order to achieve effective hydrolysis of substrates and become usable by PHA-producing microorganisms. These pretreatments improve the yield of subsequent hydrolysis by improving the accessibility to cellulose and hemicellulose due to the combined effects of delignification, reduction in cellulose crystallinity, and restructuration of the composite [58]. The pretreatments are classified as physical, physicochemical, chemical, biological, electrical, and combinations of these [55]. The advantages and drawbacks of using each pretreatment method have been reviewed elsewhere [9,56,59]. Thus, the chemical method consists of the application of acid or alkaline treatments, which renders the biomass more susceptible to enzymatic degradation [56]. Among green methods, hydrothermal treatment or autohydrolysis, also known as subcritical water extraction (SWE), has been widely tested. This consists of aqueous extractions at high temperatures and pressure, allowing for the separation of a liquid fraction rich in bioactive extractives and a solid fibrous fraction integrated mainly by cellulose, hemicellulose, and the insoluble fraction of lignin. Despite the disadvantage of using harsh temperature and pressure conditions, this process produces high sugar yields while not requiring the addition of chemicals, resulting in reduced environmental pollution [59]. The solid fraction can be subsequently submitted to further delignification by bleaching, which yields purified cellulose fibers accessible for chemical or enzymatic saccharification [60]. In any case, the conditions of all treatments prior to biosynthesis should be carefully selected because of their effect on the final hydrolysate composition and the possible production of toxic substances. Ahn et al. [61], testing different conditions for the acid digestion of rice straw waste, found that increased heating time seemed to facilitate the decomposition of sugars coupled with increasing the concentrations of inhibitors in the hydrolysates such as acetic acid, formic acid, levulinic acid, furfural, and 5-hydroxy methyl furfural.

After delignification, the hydrolysis of cellulose and hemicellulose to obtain fermentable sugars can be achieved by chemical or enzymatic protocols, often combined in consecutive steps [55]. In chemical hydrolysis, a concentrated or diluted acid is used (frequently sulphuric or hydrochloric), where the acid can penetrate lignin without pre-treatment at a fast rate [62]. Although amorphous hemicellulose requires less severe conditions to be hydrolyzed than the crystalline fraction of cellulose, a high temperature is often applied to accelerate the conversion. On the downside, this protocol promotes the decomposition of sugars, creating toxic compounds. Alkalis can also be used for chemical hydrolysis, which is believed to hydrolyze by saponification of intermolecular ester bonds crosslinking xylan hemicelluloses and other components, thus increasing the porosity of the material. Generally, alkali saccharification enhances the digestibility of the lignocellulose and reduces inhibitor formation [62].

Enzymatic hydrolysis has been claimed to lead to better quality hydrolysates than the chemical method, as a lower concentration of toxins is obtained, and therefore, these hydrolysates can be considered as superior carbon substrates for various biotechnological processes [55]. The extent of the saccharification depends on the amount of enzyme added, biomass composition, and cell wall structure [63]. Cellulases are primary enzymes for cellulose hydrolysis [57]. Several enzymes produced by microorganisms, mainly fungi but also a few bacteria, are commercially available for industrial and agricultural use [55].

As previously commented, toxins may be generated in the hydrolyzates, and therefore, their removal before PHA biosynthesis requires efficient and inexpensive solutions for the implementation of waste materials as substrates for PHA production [64]. Different strategies can be applied for hydrolysate detoxification, including physical (evaporation, membrane separation), chemical (neutralization, activated charcoal treatment), and biological (microbial treatment, enzymatic detoxification). Evaluating the toxins present and the need for removal is crucial to selecting the most appropriate method [55].

After the saccharification of the pretreated material to yield fermentable sugars, the next step would be the appropriate fermentation of sugars for PHA production [57]. An alternative process, called consolidated bioprocessing, carries out the hydrolysis and fermentation of the material in a single process [63]. This strategy involves having the two processes take place within the same bioreactor, where a single organism or a consortium is used to both produce the enzymes required for hydrolysis and perform fermentation [56]. This consolidated bioprocessing can potentially reduce the process cost significantly, but it faces some challenges, such as the cost of enzymes. Current research in metabolic engineering is focused on the design of an efficient cell factory for the direct conversion of biomass into valuable products [65]. In this sense, engineering enzyme secretomes is extending the functionality of consolidated bioprocessing, although enzymes continue to be a major cost factor [63].

## 6. PHA Producing Microorganisms: The Case of *Haloferax mediterranei*

Many microorganisms have been screened to produce PHAs, but only a few are actual candidates for commercial production [3,55], and hence, the selection of a PHA-producing strain is key to profitability. The suitability of a strain for high PHA production depends on a variety of factors: stability and safety of the organism, its growth rate and the PHA accumulation rate, the extractability of the polymer, and the molecular weight of the accumulated PHAs. Moreover, in the specific case of PHA production from lignocellulosic waste, the strain should also be able to utilize fermentable sugars present in the hydrolysates, which include pentoses, and able to tolerate or even eliminate potential microbial inhibitors [55]. Several microorganisms have been reported to produce PHBV at sufficient rate and yield for commercial production, which includes *Cupriavidus necator*, *Azotobacter vinelandii*, *Alcaligenes* spp., *Pseudomonas oleovorans* and recombinant strains of *Bacillus megaterium* and *Escherichia coli* [15] and a variety of halophile microorganisms. Koller et al. [54] reviewed the utilization of new powerful production strains, including new wild types of PHA-producing organisms, mixed microbial cultures, and genetically modified microorganisms. 

Halophiles are organisms that can grow in extremely saline environments, such as saline lakes, salt pans, and marshes [66]. Depending on their salt concentration requirements, halophiles can be classified as halotolerant (0–5% NaCl), which are able to grow in high salinity as well in the absence of a high concentration of salt, slight halophiles (2–5% NaCl), moderate halophiles (5–20% NaCl) and extreme halophiles (20–30% NaCl) [67]. These salt-tolerant microorganisms can balance the osmotic pressure of the environment and, hence, resist the denaturing effects of salts [67]. Research over the past few decades has demonstrated that halophiles can serve as commercially promising microorganisms for high PHBV yields at low production costs [68]. In such high saline concentration conditions, the chances of contamination are reduced substantially, and hence, extreme halophiles can utilize non-sterile feedstocks without the concern of contamination [1,3]. In addition, halophiles can produce PHAs growing on a variety of inexpensive carbon sources [3,4]. Another important advantage relates to the extraction of the PHA produced. The high intracellular osmotic pressure of these microorganisms causes the cells to lyse in distilled water, which significantly reduces the PHA recovery cost [3] as compared with other PHA-producing microorganisms such as *Cupriavidus necator*, *Azotobacter vinelandii*, *Alcaligenes* Spp., *Pseudomonas oleovorans,* and *Bacillus megaterium* [15,54]. In this way, the extraction of intracellular PHA does not require a high amount of chemical solvents [4], thus allowing a relatively simple downstream processing of the product [39].

Halophilic microorganisms, both bacteria and archaea, have been isolated from a range of environments [67]. Halophilic bacteria include genera such as *Halomonas*, *Halobacillus,* or *Pseudomonas*, whereas *Haloferax* or *Haloarcula* genera belong to the halophilic archaea domain [67]. Archaeals of the family *Halobacteriaceae* are examples of well-adapted and widely distributed extremely halophilic microorganisms. Although this family includes 30 genera, only some of the strains belonging to this family have been found to accumulate PHAs [16]. The genus *Haloferax* grows quickly compared with other halobacteria, with generation times in the range of 3–4 h under optimal conditions [69]. *Haloferax* is also able to use a wide range of substrates as the sole carbon and energy source, including some sugars and polysaccharides. Among them, *Haloferax mediterranei* has received great attention as the best PHA producer due to its high growth rate, metabolic versatility, and genetic stability [16,52,70]. It has been reported to accumulate as much PHA as *Ralstonia* [71], and in a comparative study carried out with different halobacteria, *Haloferax mediterranei* exhibited the highest PHB productivity. Different studies concluded that *H. mediterranei* gave the best PHBV accumulation without HV co-feeding and is the most promising candidate for industrial PHA production due to its robustness and stability potential [42,72].

*Haloferax mediterranei* is an extremely halophilic microorganism that can only grow in high-concentration saline media above 150 g/L [6]. In this environment, very few organisms can develop at growth rates anywhere near as high as those reached by *H. mediterranei* [73]. This considerably reduces the possibility of microbial contamination during cultivation and, therefore, the requirement for sterile conditions [73,74]. This property can also facilitate simplified production systems of PHA with *Haloferax mediterranei,* such as the use of open ponds similar to those used for sewage treatment [73]. Gonzalez and Winterburn [75] reported that the cultivation process of *H. mediterranei* could be operated continuously in an open vessel on a large scale. On the downside, the extreme salinity required by haloarchaea incurs higher chemical costs and accelerates the corrosion of stainless steel fermentation tanks [70]. To avoid corrosion, a high-quality steel or an inert polymeric material such as polyether ether ketone should be used [76]. Ghosh et al. [4] used artificial seawater of 3.7% total salinity to produce PHA from macroalgae hydrolysates. With this approach, using seawater seems a viable strategy for PHA production. Montemurro et al. [77] used wasted bread as a carbon substrate for microbial growth, to which microfiltered seawater was added instead of the expensive minerals supplement frequently used for this microorganism. 

*Haloferax mediterranei* was first reported to produce PHB [78]. Later on, it was found that it can naturally produce PHBV from carbon sources such as glucose, starch, or other inexpensive industrial by-products without any supplementation of 3HV precursors [1,16,54,69]. Given that HV precursors such as propionic and valeric acids are often toxic to the cells, the natural accumulation of PHBV from structurally unrelated carbon sources represents an advantage of *Haloferax mediterranei* over other PHA-producing microorganisms [39,79]. Thus, *Haloferax mediterranei* can grow on glycerol from the biodiesel industry [80] as well as agricultural and industrial wastes such as whey, rice bran, vinasse, and stillage [39]. This fact could be related to *H. mediterranei’s* ability to use pentose sugars, which are generally more difficult to ferment than their hexose counterparts [81]. Thus, *H. mediterranei* has been reported to utilize xylose and arabinose to yield PHA contents of 33.4% w/w and 17.34% w/w, respectively [4]. Cellobiose and soluble starch have also been reported as usable carbon sources for *Haloferax mediterranei* [73,82]. Table 2 provides an overview of recent studies on PHA production by *Haloferax mediterranei* using inexpensive substrates coming from waste food stocks as carbon source, where some parameters related to the efficiency of the process are summarized: the maximum concentration of cell dry mass (g CDM/L) and PHA (g PHA/L), the percentual accumulation of PHA in the cells (g PHA/100 g CDM), the productivity yield (g PHA/g carbon source), and the PHA volumetric productivity (mass of PHA per volume and time). 

Some additional advantages have been reported on the use of *H. mediterranei* for PHA production. On the one hand, its genome sequence is known, and therefore, molecular biotechnology can be applied to PHA synthesis processes [83,84]. In this sense, Zhao et al. [85] succeeded in the genetic modification of the archaeon, significantly enhancing the PHBV production (by about 20%). Moreover, *H. mediterranei* has been shown to produce high-value substances concurrently with PHA, such as an extracellular polysaccharide, which can be used in food technology, bacterioruberins that are widely applied as food additives and medicinals [39], and carotenoids [52]. The commercial exploitation of such substances may reduce the overall cost of PHA production. Furthermore, *H. mediterranei* does not produce lipopolysaccharide endotoxins, contrary to what happens when using gram-negative bacteria. This is an advantage because these toxins may co-purify with PHA, restricting its biomedical applications [39,70].

**Table 2 foods-13-00950-t002:** Recent studies on PHA production by *Haloferax mediterranei* using waste materials as carbon sources.

Reference	Substrate	gCDM/L	gPHA/100 gCDM ^(1)^	g PHA/L	Product Yield Coefficient ^(2)^ (Y_P/S_)	Volumetric Productivity	Other Findings
[4]	Simulants of green macroalgae hydrolysates	3.8 ± 0.2	58%	2.2 ± 0.12 g/L	---	0.035 g/L h	Conditions for fermentation are to be optimized to increase the yield and productivity of PHA production.
[6]	Sesame seed waste water hydrolysates (acid hydrolysis)	50	75%	0.53 g/L to 20.9 g/L with glucose supplementation	---	---	Less than 1 g sugar/L hydrolysate. Glucose supplementation needed.
[80]	Biodiesel industry-derived by-products	---	75.4%	16.2 g/L from crude glycerol phase	0.19 g/g	0.12 g/L h	10% mol HV, 253 kDa polydispersity 2.7
[86]	Olive mill wastewater	---	43%	Max 0.2 g/L	---	---	6.5% mol HV
[87]	Synthetic molasses wastewater	6	---	5 g/L	---	390–620 mg/L h	Increased T (35 °C) improves production
[52]	Chlorella biomass hydrolysates (acid hydrolysis)	6	55.5%	4 g/L	---	---	PHBV with 10.5% mol HV
[1]	Food Waste-Derived Nutrients	3	---	2 g/L	0.41–0.54 gPHBV/g acetate	---	Up to 23% wt. HV
[88]	Fermented food waste	7.0 ± 0.7	---	4.5 ± 0.2 g/L PHA	0.3 g PHBV/g COD	---	10–30% wt. HV
[89]	Ricotta cheese exhausted whey enzymatically treated	6 (lab), 18 (pilot)	13.5% lab 9% pilot	1.0 g/L lab 1.3 g/L pilot	0.05–0.14	0.05–0.21 g/L h	Approx 16% HV
[16]	Vinasse from etanol industry	---	70%	19.7 g/L	0.87 (based on total carbohydrates)	0.21 g/L h (based on total carbohydrates)	PHBV 12–14 mol%
[83]	Rice-based ethanol stillage		71%	16.42 g/L	0.35	0.17 g/L h	PHBV 15.4% HV
[39]	Stillage from rice-based ethanol manufacture		63%	13.12 g/L	0.27	0.14 g/L h	PHBV 17.9% HV
[78]	Chemically hydrolyzed cheese whey	7.54	53%	7.92 g/L	---	4.04 g/L day	1.5 mol% HV. The polymer presented a molecular mass of 4.4 × 10^5^, with a polydispersity index of 1.5.
[90]	Extruded rice bran and extruded cornstarch	140	55.6%	77.8 g/L	---	---	Repeated fed-batch fermentation
[72]	Whey lactose	---	73% PHA 50% PHBV	---	0.29	0.09 g/L h	*H. mediterranei* was a more effective producer than *Pseudomonas hydrogenovora* and *Hydrogenophaga pseudoflava*
[76]	Whey lactose	---	50%	5.5 g/L	---	0.05 g/L h	---
[77]	Waste bread	2–3	---	1.53 g/L	20–25 mg/g, depending on the extraction procedure	---	10–13% (w/w) HV depending on the extraction procedure. Solvent-free extraction could be used
[13]	Silkworm excrement	---	---	1.73 g/L	---	---	16% mol HV
[91]	Hydrolyzed rapeseed meal	5–10	---	0.512 g/L	---	---	---

^(1)^ CDM: cell dry mass. ^(2)^ Yield: mass of PHA referred to the mass of substrate.

Table 1 shows that both the percentual accumulation of PHA in the cells and the HV content in PHBV are very variable. The accumulation of the polymer starts during the logarithmic phase and increases as the biomass increases with some delay [73]. PHA synthesis is triggered by different environmental stimuli, such as nitrogen or phosphorous limitation. Given the effect of culture conditions and nutrient supplies on the accumulation of PHA, this can vary widely. *H. mediterranei* is capable of producing more than 70% PHA, but much lower percentages are also reported since culture conditions and nutrients seriously affect the PHA accumulation capacity of the microorganism [77]. 

As regards the yield of PHA with respect to the carbon source, *Haloferax mediterranei* was reported to present a very high yield value in comparison with other microorganisms. Lillo and Rodríguez-Valera [73] commented that the maximum yield of this archaeon is comparable to that of *A. eutrophus* under optimal conditions (0.33 g/g), which is remarkable considering the fact that the archaeon produces an extracellular polysaccharide as well as the PHA.

The cultivation technique affects the biopolymer titer and productivity [92]. PHAs can be produced in shake flasks and in bioreactors, where different conditions are fixed [37]. PHA production experiments at a laboratory scale are often performed in shake flasks, with the aim of exploring the ability of a microorganism to produce the biopolymer. Usually, batch production does not result in high productivity because the initial substrate concentration is limited to the growth of the organism [92]. For this reason, a fed-batch strategy is often preferred, where a supplement of the substrate is added to the reactor at a specific rate from a reservoir as biosynthesis progresses. Thus, consistent supplementation of the growth substrate was often carried out for improved PHA production [77]. Shake flask experiments may be performed prior to fed-batch bioreactor tests. Priya et al. [15] carried out shake flask fermentations at varying nutrient concentrations to optimize the cultivation conditions, after which *H. mediterranei* was cultivated for PHBV production in a fed-batch bioreactor with pulse-fed levulinic acid. The results found in this study can serve as a reference for future large-scale commercial production of PHA. Even though the most reported fermentation modes for PHA production with *Haloferax mediterranei* are batch or fed-batch, these strategies have been reported to give a low quantity of PHA of variable quality [6]. Aiming to increase PHA productivity with the desired composition and minimal variations in product properties, continuous PHA fermentation has been considered as a potential alternative [6]. Provided that contamination is avoided and the stability of the strain is guaranteed, continuous culturing offers many advantages for industrial production, such as simplicity of culture control, homogeneity of the production, and constancy of culturing conditions [73].

The potential for the valorization of food waste underscores the importance of research on the scaling-up of these processes from laboratory to industrial scale [15], as upscaling to industrial scales faces numerous challenges [93]. Bhattacharyya et al. [39] scaled up their previous laboratory-scale bioreactor study where *Haloferax mediterranei* was used to produce PHA using rice-based ethanol stillage and succeeded in producing very pure PHBV. Raho et al. [89] tested a PHA production process using ricotta cheese exhausted whey as a carbon source up to pilot scale conditions and found that both CDM and polymer synthesis were markedly higher compared with the in-flasks trials, even though the yield was lower. Wang et al. [68] designed and analyzed an industrial-scale PHA production system using *H. mediterranei* and cheese byproducts as feedstock and concluded that utilizing dairy-derived feedstocks has the potential to make PHA competitive in the bioplastic market. Gonzalez and Winterburn [75] tested a continuous feeding of volatile acids strategy, an alternative to the usual pulse-fed protocol used in fed-batch fermentations. It was found that this novel feeding strategy caused a significant increase (*p* < 0.05) in productivity and yield while retaining control of the polymer composition.

PHA downstream processing plays a vital role both for material quality and cost of production [49]. The first step in downstream processing is the separation of cells via centrifugation, filtration, or sedimentation [92]. As commented on above, generally for halophilic microorganisms, the extraction of PHAs from *Haloferax mediterranei* is facilitated by osmotic shock caused by hypotonic environments and leads to cell lysis, which may require less chemicals and energy than other PHAs production systems [1,4]. Montemurro et al. [77] tested a solvent-free extraction process, where no purification step with organic solvent was applied, obtaining a 93% purity of PHBV with 12% HV content and concluding that an extraction process exclusively based on osmotic shock could be used to recover the bioplastic from cells efficiently. 

After fermentative production of PHA-rich biomass, cell harvest, and downstream processing for PHA recovery, highly saline waste streams (namely exhausted medium broth and cell debris) remain as process residues. Once the fermentation is concluded, a complex mixture of compounds remains in the cultivation medium, which includes leftover nutrients not fully consumed (such as residual sugars, short-chain carboxylates, and micronutrients) as well as cell metabolites generated during cell growth (such as acetic acid and extracellular polysaccharides. Aiming for clean and sustainable production of PHAs, zero-emission of spent fermentation broth and saline wastewater should be attained [84]. Strategies to close the saline wastewater cycle, which includes the recyclability of the spent fermentation broth and the conversion of a salt-rich fraction of biomass remaining after product recovery, need to be assessed [84].

The recycling of spent saline media after a batch of fermentation can provide both environmental and economic benefits to the whole PHA production process by saving the purchase of salts and the expensive treatment of high saline wastewater. If multiple recycled batches are performed, the impact of these compounds on cell growth and production yield should be thoroughly studied. Wang et al. [88] successfully tested the efficacy of chemical oxidation by H_2_O_2_ (frequently used in industrial wastewater treatment), followed by rotary evaporation, in removing excessive chemical oxygen demand from the spent saline medium. In their study, consistent PHA concentration and polymer quality were achieved for four consecutive batches. Koller [84] reported that spent fermentation broth can be used to replace a part of fresh fermentation medium in subsequent cultivations and that 29% of yeast extract used as nitrogen and phosphate source can be replaced by cell debris from prior production processes. Bhattacharyya et al. [83] reported on a desalinization method achieving the recovery of 96% of the medium salts for re-use, hence allowing them to meet the environmental standards of total dissolved solids in discharge water. These results on desalinization and the successful recycling of both spent media and cell debris are both of economic and environmental significance, given the environmental risks connected to the disposal of salty waste streams.

## 7. Strategies for Improving the Economic Viability of PHAs Production and Profitability: Packaging and Other Applications

In addition to using non-expensive carbon sources such as agrifood waste, other measures can be taken to improve the profitability and, hence, the viability of PHA production and application. The possibility of constructing a microbe capable of profitably synthesizing products with economic feasibility follows from the advent of recombineering DNA and CRISPR technologies [94]. Along with the favorable accumulation of PHA, this strategy may promote the synthesis of quite a few high-market value chemicals, which are concurrently produced with PHA. Even wild-type strains of *Haloferax mediterranei* have been reported to produce high-market value products such as extracellular polysaccharides with thickening power, microbial proteins, carotenoids, archaeal lipids, and bacteriocins [84,95,96]. The use of integrative approaches to exploit the production of multiple concomitant bioproducts with added value from PHA synthesis contributes to the enhanced economic feasibility of the whole process.

*Haloferax mediterranei* has been reported to produce an exocellular polymeric substance (EPS), a heteropolysaccharide containing sugars, aminosugars, uronic acids, and sulfate, in yields as high as 3 mg/mL, which gives the colonies its mucous character [97]. Although potential applications of this polymer in the food industry have been pointed out [71,96], it should also be noted that EPS excretion complicates the biotechnological cultivation process of *H. mediterranei* [84], by diversion of part of the carbon flow from PHA synthesis [1]. The potential of excreted EPS for neutraceutical, pharmaceutical, and therapeutic applications needs further research efforts before the pros and cons of its production, as compared with that of PHA, are determined [84].

Carotenoids have received considerable attention recently because of their potential benefits for human health. Accordingly, the interest in the production of natural carotenoids by microbial fermentation has increased [98]. *Haloferax mediterranei* is able to produce C50 (bisanhydrobacterioruberin, monoanhydrobacterioruberin, and bacterioruberin) and C45 (2-isopentenyl-3,4-dehydrorhodopin) carotenoids which have different functions such as fluidity of the membrane, adaptation to salinity and temperature, and scavengings of free radicals [84]. C50 carotenoids have remarkable antioxidant capacity due to their high number of conjugated double bonds, which makes these carotenoids very interesting for the food and pharmaceutical industries. Possible applications of haloarchaeal carotenoids include potential uses in biomedicine as antitumoral, antiviral, or spermatic mobility enhancers, as well as in the prevention of skin cancer [99].

Bacteriocins are biologically active proteins of bacterial origin exhibiting antimicrobial properties against other microorganisms, generally closely related to the producer. These protein antibiotics have been studied due to their capability as food preservatives [100]. Halocins, bacteriocins produced by haloarchaea, can be used for the preservation of salted food products and the preservation of hides in the leather industry. They also have clinical applications such as the prevention of cardiac injury and anti-cancer activity [101]. The main drawback associated with the separation of marketable chemicals or fractions over PHA biosynthesis is the effective separation from extractives, given the usual low concentrations of these valuable products in the process streams. However, economic profitability can be reached in specific cases oriented to high-added-value products [60].

Other strategies deal with the optimization of production conditions. In this sense, the correct selection of nitrogen source and temperature results in reduced costs. The nitrogen source for PHA biosynthesis attains a significant share of the production cost, and therefore, the proper selection of inexpensive nitrogen sources such as urea and sodium nitrate reduces the cost of production significantly [37]. On the other hand, energy requirements to maintain constant temperature over fermentation should also be taken into account. The best temperature for *Haloferax mediterranei* growth has been reported to be 45 °C [69], but many studies set it at 37 °C on the basis of two advantages: saving energy costs over long-term continuous culture and saving oxygen cost in high cell density culture (lower temperature promotes a higher saturated oxygen concentration) [90]). Cui et al. [87] studied the effect of temperature on PHA production by *H. mediterranei* and found that high temperature (35 °C) resulted in fast metabolism, triggering an increased PHA storage even without nitrogen limitation. Considering the increased cost that heating would imply for PHA production and the usual warm temperature of wastewater, actions should be taken to maintain this condition during transport and treatment.

PHA-based materials exhibit excellent barrier properties toward oxygen, carbon dioxide, and moisture, thus making them suitable for food packaging [102]. The barrier and mechanical properties are similar to those of conventional non-biodegradable plastics frequently used, such as PP, PE, and PET. PHB, PHBV, or poly(3-hydroxyvalerate) (P(3HV) or PHV), are thermoplastic polyesters that could replace petroleum-based polymers in most packaging applications, with the food sector being one of the main targets [17]. Moreover, PHA applications in the packaging industry have also been related to developing cosmetics containers, shampoo bottles, shopping bags, and cups [103,104].

Despite the acceptable results regarding the impact on the food quality of PHB-based materials, its application as food packaging has been delayed because of its brittleness, stiffness, thermal instability, limited gas barrier properties, as well as high cost and limited availability [17,30]. In this sense, different strategies have been investigated to improve PHA functional properties, such as packaging material, and extend their applicability. Copolymerizing approaches have been effective alternative means of improving the thermal stability, mechanical performance, and gas and water vapor barrier capacity of these biopolymers by decreasing both their brittleness and melting temperatures [17,30,105,106,107]. Several copolymers have been developed using different monomer units, such as 3-hydroxyvalerate, 4-hydroxybutyrate, or 3-hydrohexanoate, thus improving the physical properties of the PHB homopolymer [108]. PHBV obtained by different levels of copolymerization of 3-hydroxyvalerate leads to less stiff, tougher materials with improved thermal stability and barrier properties, with a broadened application range in the food packaging sector [30,32,105,106,107].

The incorporation of different fillers from low-cost feedstocks, such as natural fibers coming from lignocellulosic residues, previously hydrophobized for improved compatibilization with PHAs, could reduce the final cost of the PHA-based composites. This strategy would, in turn, contribute to the revalorization of agri-food by-products for more sustainable and environmentally safe packaging materials at low cost. Although organic fillers and fibers composed of cellulosic material can improve the properties of polymers, their effect on the marine biodegradability of the composite remains unexplored [109]. Apart from cellulose, other cheaper biobased fillers, such as proteins and starch, have been reported to improve biodegradability rates compared with other fillers [109]. Many possible fillers and fibers could be considered, and their effect on the final properties of the PHA-based composite subjected to further research.

Blending with other less expensive biodegradable polymers could be a good strategy both to reduce the final cost of the blend material and to improve its properties. Previous research has reported the improved properties of PHA blends. Requena et al. [110] obtained compression-molded PLA:PHBV blend films with improved extensibility and water barrier capacity. Combinations of PLA-PHB blends have been functionalized with cellulose nanocrystals for use in short-term food packaging [111]. Multilayer systems, where different materials are combined in layers, constitute an alternative approach. Soares da silva [112] combined a bacterial nanocellulose-PHBV layered composite for food packaging with complementary barrier properties. Electrospinning also constitutes a promising technique for the development of PHA-based food packaging materials [113,114,115]. 

Research efforts are being made to increase the added-value of PHA-based materials by designing active biodegradable systems for food packaging applications as a follow-up of biomedical applications of PHA-antimicrobials systems. Biodegradable materials exerting antimicrobial/antioxidant properties represent innovative food packaging alternatives because they have a positive impact on the food spoilage processes, thus extending shelf-life [116,117,118]. These agents are of varied origin, including plant (essential oils, among others), mineral (transition metals and metalloids), animal (chitosan, propolis), and microbial (bacteriocines such as nisin or pediocin). In this sense, recent studies report on the effect of different antimicrobial agents incorporated into PHA-based matrices, such as vanillin [119], eugenol [120], carvacrol [121]), phenolic acids [122], silver nanoparticles [123], biogenic silica [124], or quercetin [125]. A promising research line consists of the incorporation of antimicrobial extracts from food waste with proven antimicrobial properties into PHA matrices for food packaging applications. Zare et al. [126] incorporated an antimicrobial plant extract into PHBV-chitosan systems and reported on their efficacy in extending the shelf-life of poultry products.

Antimicrobial incorporation into biopolymer matrices may affect their overall properties as food packaging materials, and hence, the effect of this incorporation on the material properties should be studied. Some antimicrobials can improve PHA-based systems in terms of their thermostability, mechanical behavior, and barrier properties [8]. For example, Castro-Mayorga et al. [123] found that incorporating only 0.04 wt % of Ag NPs into PHBV reduced 56% oxygen permeability compared with neat PHBV. Romero-Castelán et al. [8] found that PHAs protect natural antimicrobial compounds by increasing their degradation temperature and thus their loss overheating, being the release and activity of the antimicrobial compounds prolonged when they are trapped within PHAs. Even volatile natural antimicrobial compounds remained active longer than their free forms when entrapped in a macromolecular matrix of PHAs [8]. Bonnenfant et al. [125] studied the stability under reuse conditions of PHBV and PHBV-quercetin systems and found small changes in the intrinsic properties of the material. 

Antimicrobials may also affect the biodegradability kinetics of the final material. These changes are attributed to the influence of the antimicrobial compounds on the microorganism population responsible for the biodegradation process [127].

## 8. Conclusions and Future Challenges

In this work, an overview of the state of the art, as regards PHA production, has been given, with a special focus on some of the available strategies that are being explored for improving the process profitability. The utilization of lignocellulosic waste as an inexpensive carbon source appears to be a key strategy in this process and entails the need for pretreatments to improve saccharification. Furthermore, the whole process can be potentially simplified by combining hydrolysis and biosynthesis in a unique step. In this sense, consolidated bioprocessing seems to be a promising alternative for PHA production but further studies are needed to understand its potential as well as to find ways to make the process feasible. 

Halophilic microorganisms, and especially *Haloferax mediterranei*, are promising producers of PHAs, presenting some relevant features that make them especially attractive, such as their ability to grow in non-sterile conditions and the possibility of using seawater for the preparation of the medium. Additionally, downstream processing for PHA extraction obtained from *H. mediterranei* can be simplified by treating the cells with pure water, which would cause hypo-osmotic shock, thus decreasing the cost of operation. Further research should focus on the optimization of recycling conditions for the effluent treatments on a pilot to commercial production scale and the impact of sidestream reutilization and desalinization economic viability as an integrated part of the PHA production process. As most research on PHA production has been performed at a laboratory scale, future studies should focus on the up-scaling of the process, aiming to assess the properties of the biopolymer obtained. 

The packaging materials obtained from polydydroxyalkanoates are considered a promising alternative to conventional plastic due to their biodegradability, biocompatibility, and good physicochemical properties, especially for medical and food packaging applications. They exhibit good mechanical and barrier properties, similar to those of conventional non-biodegradable plastics frequently used for this purpose. The improvement of the properties of the material is an active area of research, given that its reduced flexibility limits its utilization in food packaging. Thus, some key strategies are being studied, such as blending with other polymers to reduce brittleness and nanofiller inclusion to increase toughness. Furthermore, the incorporation of antimicrobial and/or antioxidant compounds in the biopolymer matrix to design active materials with an appropriate release capacity is being explored. 

As a final remark, research efforts on different fronts are still needed to improve the economic viability for profitable PHA production at a large industrial scale. Further improvements could also allow for the production of even more flexible grades of PHAs or transparent ones through the control of its crystallization, which will promote its extended use as bioplastic for active packaging applications.

## Figures and Tables

**Figure 1 foods-13-00950-f001:**
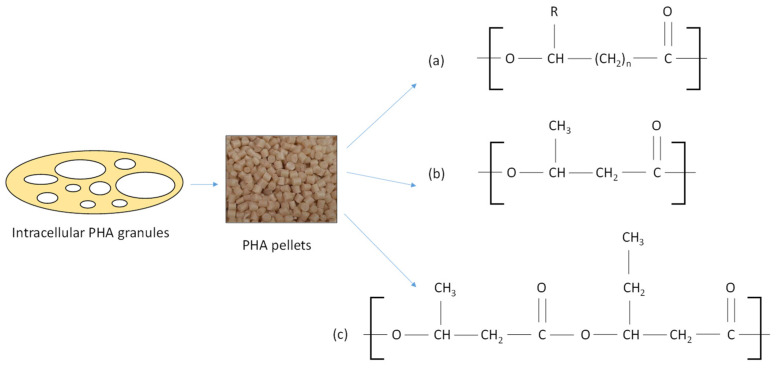
PHA pellets obtained from intracellular granules and the general structure of PHAs (**a**), the structure of PHB (**b**), and the structure of PHBV (**c**).

**Figure 2 foods-13-00950-f002:**
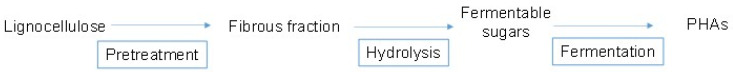
General procedure for PHA production from lignocellulosic feedstocks.

**Table 1 foods-13-00950-t001:** Properties of packaging films based on polyhydroxyalkanoates (PHAs).

Type of PHAs	Production Technique	T_g_ (°C)	T_m_ (°C)	EM (GPa)	TS (MPa)	E (%)	X_c_ (%)	WVP·10^15^	OP·10^19^	Reference
PHB	Solvent casting	−2–9	160–180	1.2–3.6	19–44	2–5	57–63	2–10	8	[20,21,22]
	Melt blending	10–18	170–180	1.5–3.5	8–40	0.8–2.1	45–50	8	5–8	[23,24,25]
P(3HB-3HV_8_ *)	Melt blending	7	170.6–171.2	1.7	37	3.4	73.8–99.06	5.7	-	[26,27]
P(3HB-3HV_12_ *)	Solvent casting	-	140–155	0.6	6.4	1.4	33	50	15	[28]
	Melt blending	−3–2	140–155	1.7	14.6	1.2	37	4	15–18	[29,30,31]
P(3HB-3HV_50–55_ *)	Solvent casting	−10	77	-	16	1200	-	-	-	[29]
	Melt blending	0	162	0.4	13.4	230	-	-	-	

T_g_: Glass transition temperature; T_m_: Melting temperature; EM: Elastic modulus; TS: Tensile strength; E: Elongation at break; X_c_: Degree of crystallinity; WVP: Water vapor permeability (kg·m/m^2^·Pa·s); OP: Oxygen permeability (m^3^·m/m^2^·s·Pa). * Indicates the mol% comonomer unit.

## Data Availability

The original contributions presented in the study are included in the article, further inquiries can be directed to the corresponding author.

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
