# Peer review of "Production of Polyhydroxyalkanoates for Biodegradable Food Packaging Applications Using Haloferax mediterranei and Agrifood Wastes"

_foods, 2024, doi:10.3390/foods13060950_

Round 1

Reviewer 1 Report

Comments and Suggestions for Authors

 Production of polyhydroxyalkanoates for active food packaging

 applications using Haloferax mediterranei and agrifood wastes

Foods

 Comments

This interesting review describes extensively the production of PHAs and suggests the use of Haloferax mediterranei as effective alternative to obtaining the products more easily and with higher yields. At least, one colored figure describing the process should be included in the manuscript. The manuscript needs minor corrections.

Specific comments

P2, L51 – PLA is not a microbial-based polymer! PLA is biodegradable (not under aquatic environments, correctly written in this manuscript), but can be produced by two synthetic routes. Please, correct.

P6, L258 – What is 5-HMF? Explain in the text.

P6, L267 – Please, correct the use of “who” in the sentence.

P15, L638 – “... pretreatments for to ...”. Please, correct.

P16-19, L687-888 – References must be standardized in accordance with the Guidelines for authors:

  • Journal Articles:
    1. Author 1, A.B.; Author 2, C.D. Title of the article. 
    Abbreviated Journal Name YearVolume, page range.
Comments on the Quality of English Language

The quality of the English Language is very good.

Author Response

P2, L51 – PLA is not a microbial-based polymer! PLA is biodegradable (not under aquatic environments, correctly written in this manuscript), but can be produced by two synthetic routes. Please, correct.

Response: Thank you for your comment. It has been corrected. We have used the term biobased instead, which also applies to PLA.

P6, L258 – What is 5-HMF? Explain in the text.

Response: It is hydroxy methyl furfural. It has been explained in the text.

P6, L267 – Please, correct the use of “who” in the sentence.

Response: It has been corrected.

P15, L638 – “... pretreatments for to ...”. Please, correct.

Response: It has been corrected.

P16-19, L687-888 – References must be standardized in accordance with the Guidelines for authors:

  • Journal Articles:
    1. Author 1, A.B.; Author 2, C.D. Title of the article. Abbreviated Journal Name YearVolume, page range.

Response: All referenes have been standarized.

Reviewer 2 Report

Comments and Suggestions for Authors

This paper describes in detail the methods and processes for the production of PHA using the Haloferax mediterranei and inexpensive agrifood wastes. Some suggestions are also made on the problems of PHA production and how to reduce its production cost. The article is detailed and clear, but there are some problems.

1. In Introduction, 1.1, 1.2 and 1.3 should be placed as a separate chapter in chapter 2.

2. There are a few detail errors. For example, "c" is not marked in Figure 1, please add the label.

3. In line 220, "Figure 1" should be corrected as "Figure 2".

4. The text of the application of PHA in food packaging is less, it is recommended in Chapter 4 to increase the application of PHA in food packaging examples.

5. The text should be specifically listed PHA can be used for the preparation of which food packaging materials, what are the deficiencies of these materials, and give the corresponding recommendations.

6. The degradation mechanism of PHA is not mentioned in the text, please describe briefly how PHA is degraded.

Author Response

  1. In Introduction, 1.1, 1.2 and 1.3 should be placed as a separate chapter in chapter 2.

Response: It has been changed in the revised manuscript.

  1. There are a few detail errors. For example, "c" is not marked in Figure 1, please add the label.

Response: Figure 1 has changed according to reviewer’s 1 comments.

  1. In line 220, "Figure 1" should be corrected as "Figure 2".

Response: It has been corrected.

  1. The text of the application of PHA in food packaging is less, it is recommended in Chapter 4 to increase the application of PHA in food packaging examples.

Response: More examples on the application of PHAs to food packing have been added (lines 682-686; 698-699; 701-703; 715-717).  

  1. The text should be specifically listed PHA can be used for the preparation of which food packaging materials, what are the deficiencies of these materials, and give the corresponding recommendations.

Response: This information has been included in the revised manuscript following the reviewer’s comment (lines 645-663).

  1. The degradation mechanism of PHA is not mentioned in the text, please describe briefly how PHA is degraded.

Response: PHAs degradation mechanism has been described (lines 164-171).

Reviewer 3 Report

Comments and Suggestions for Authors

Dear authors,

I would like to acknowledge interesting topic that you have chosen for your work. First part of the paper is very well written, widely referenced and sound. However, this is the part of the paper that has already been reviewed in the near past: PHA production using agri waste streams, as well as halophile organisms use in the PHAs production. The third part of the paper, that has not been reviewed recently, food packaging applications and especially active food packaging application is unfortunatelly given very scarce, only as a note.

My main recommendation due to the above is to give through review of the current trends and research in the field of PHAs use for food and active food packaging applications, important material properties and effects, or to change the article title, because it is misleading as it is.

In case you choose not to further deepen the packaging application section, please give explicit statement of what is new in the review that has not been already reported in the last few years.

Author Response

Response: Following the reviewer’s comment, we have changed the title of the paper and we have added more information and a new table on the properties of PHAs based packaging materials (lines 114-162 and Table 1). In addition, more examples on the application of PHAs in food packaging have been included (lines 645-663; lines 682-686; 698-699; 701-703; 715-717). 

Reviewer 4 Report

Comments and Suggestions for Authors

The current review article is centered on the production of polyhydroxyalkanoates from agri-food waste as well as their potential applications as sustainable packaging materials. The review article is succinct, providing timely and relevant insights with respect to the general benefits of PHA as a potential alternative to fossil derived packaging materials, the challenges associated with it's industrial production/widespread application and potential strategies to achieve scalability and competitiveness vis-a-vis traditional plastic packaging materials. In addition to these points, there are a number of observations which the authors are encouraged to consider.

-In the Title, active packaging should be replaced with either sustainable packaging or biodegradable packaging since the review article did not provide any substantive discussion on active packaging.

-In the Abstract Line 7, please PHA should be written in ful since this the first time in which the term is being used.

--Lines 7-11 as a single sentence is too long which hinders understanding. To ensure clarity and facilitate comprehension, authors are encouraged to split up the sentence into smaller phrases.

-Although the Abstract provides a general picture of what the review article was about, it would be more meaningful for more details to be included to give it more depth.

-With respect to the 'structure and properties of PHA' it is expected that the properties of PHA be discussed more robustly in comparison with those of the mentioned fossil-based plastic materials. Please provide specific examples and numerical values of the said properties.

-The authors should subject the manuscript to rigorous English language editing to rid it of the many errors.

-Line 95: monomeric unit or monomers

--Lines 199-200: It it the availability of cheap substrate or lack thereof that limits the large scale production of PHA?

-Figure 1 should be improved to increase the visual appeal. Representative images or graphics should be inserted.

Comments on the Quality of English Language

Moderate editing of English language is strongly suggested.

Author Response

-In the Title, active packaging should be replaced with either sustainable packaging or biodegradable packaging since the review article did not provide any substantive discussion on active packaging.

Response: It has been replaced.

-In the Abstract Line 7, please PHA should be written in ful since this the first time in which the term is being used.

Response: It has been written in the full form.  

--Lines 7-11 as a single sentence is too long which hinders understanding. To ensure clarity and facilitate comprehension, authors are encouraged to split up the sentence into smaller phrases.

Response: This text has been rewritten in smaller phrases.

-Although the Abstract provides a general picture of what the review article was about, it would be more meaningful for more details to be included to give it more depth.

Response: The abstract has been rewritten and more details have been included.

-With respect to the 'structure and properties of PHA' it is expected that the properties of PHA be discussed more robustly in comparison with those of the mentioned fossil-based plastic materials. Please provide specific examples and numerical values of the said properties.

Response: More specific examples and numerical values have been included (lines 114-162  and Table 1)

-The authors should subject the manuscript to rigorous English language editing to rid it of the many errors.

Response: the English language has been revised in detail.

-Line 95: monomeric unit or monomers

Response: It has been corrected.

--Lines 199-200: It it the availability of cheap substrate or lack thereof that limits the large scale production of PHA?

Response: As explained in the manuscript, the large-scale production of PHA is hindered by multiple factors among which the utilization of cheap substrates represents a good strategy for cost reduction. Their large-scale production is not a problem itself, because they are already available as subproducts of many crops. Rather, the availability of industrial-scale amounts in the proximity of potential PHA production plants or the transportation logistics may be the limiting factors in this respect.

-Figure 1 should be improved to increase the visual appeal. Representative images or graphics should be inserted.

Response: Figure 1 has been improved.